# Fishing with Pesticides Affects River Fisheries and Community Health in the Indio Maíz Biological Reserve, Nicaragua

Joel T. Betts [1,2,*], Juan F. Mendoza Espinoza [1,3], Armando J. Dans [1,2,3], Christopher A. Jordan [1], Joshua L. Mayer [4] and Gerald R. Urquhart [2,5]

[1]  Global Wildlife Conservation, Austin, TX 78746, USA; hello.jaguar.llu@gmail.com (J.F.M.E.); arjadans@gmail.com (A.J.D.); cjordan@globalwildlife.org (C.A.J.)

[2]  Department of Fisheries and Wildlife, College of Agriculture and Natural Resources, Michigan State University, East Lansing, MI 48824, USA; urquhart@msu.edu

[3]  Instituto de Recursos Naturales, Medio Ambiente y Desarrollo Sostenible (IREMADES), Universidad de las Regiones Autónomas de la Costa Caribe Nicaragüense (URACCAN), Bluefields 81000, Nicaragua

[4]  Department of Anthropology, University of California, Los Angeles, CA 90095, USA; joshuamayer@ucla.edu

[5]  Lyman Briggs College, Michigan State University, East Lansing, MI 48825, USA

*  Correspondence: joel.t.betts@gmail.com; Tel.: +1-(616)-916-7291

**Abstract:** The practice of harvesting fish and crustaceans by using pesticides is understudied and under-reported in tropical inland fisheries yet poses a significant threat to freshwater biodiversity and community health. This research provides a brief review of the practice and an in-depth case study from southeast Nicaragua. In 2019, 86 interviews and 5 focus groups were conducted in remote communities in the Indio Maíz Biological Reserve (IMBR) and nearby surrounding area and combined with 4 years of local Indigenous Rama and Afrodescendent Kriol community forest ranger data. Forest rangers and 74% of interviewees reported that fishing with pesticides occurs in their communities, including both inside the IMBR and in the nearby surrounding area. The practice is primarily used by illegal settlers, and not by Rama and Kriol communities who have rights to the land in the IMBR. It entails the release of liquid pesticides in water or mixing powdered pesticides with corn flour and using the mixture as bait. Of seven chemicals reported, Cypermethrin, Deltamethrin, and Aluminum Phosphide were most common. The use of ichthyotoxic woody plants was more rarely reported. Habitats targeted ranged from swift headwaters to slow pools in small creeks to larger rivers, depending on target species. Main uses reported for the catch were food for family, bait to catch larger fish, and for sale. The main motivation was increased catch efficiency. Many interviewees attributed stomach issues, diarrhea, cough, convulsions, and miscarriage to exposure to poisoned river water. Twenty-five interviewees blamed poisoned rivers for livestock miscarriages or death. Severe local losses of fish and shrimp populations were reported. Rama and Kriol interviewees describe the practice as a threat to their river-based food security. Despite its illegality, no study participant knew a case of pesticide fishing that had been prosecuted. This destructive fishing practice has significant implications for conservation of the intact river systems of the primary rainforests of southeast Nicaragua, and to the local traditional fisheries they support.

**Keywords:** coastal; disturbance; invertebrates; fish; fishing; pesticide; pollution; protected areas; stream; toxicity

## 1. Introduction

Pesticide exposure has long been known to cause problems for ecosystem and human health [1,2]. The specific ecological impacts for the wide variety of chemicals used in agriculture are well-studied [3]. For example, in aquatic systems, it is well known that pesticides generally reduce the abundance and diversity of fish and aquatic invertebrates and cause changes in community composition in rivers that receive polluted runoff [4,5]. In extreme cases, pesticide runoff from agriculture can result in large fish kills [6]. These effects are manifested in reduced inland fisheries' potential [7], and therefore have potential to affect the billions of people who rely on inland fisheries for food and the millions who rely on them for their livelihood [8]. Acute pesticide poisonings in people from agricultural exposure have been reported as a major problem throughout the tropics [9]. Long-term exposure to many different pesticides has been linked to cancer, nerve damage, and other acute and long-term ailments in people [10]. Regulations for standard pesticide values permitted in soil, drinking water, and agricultural commodities vary widely between jurisdictions across the world, in many cases exceeding international guidelines from the World Health Organization (WHO), and thus are often not low enough to avoid human health risks [11]. As impacts have become more understood, persistent and noxious chemicals like dichloro-diphenyl-trichloroethane (DDT), toxaphene, and many others have successfully been taken off the market and replaced by others that are less persistent in the natural environment and have fewer long-term health risks. However, in many cases these less persistent chemicals can be just as acutely dangerous to people and ecosystems in the short term. In addition, in some cases toxic residues of these less persistent chemicals can be detected in increasing concentrations in surface water over the years [3].

Although most reported ecological and human health impacts from pesticides are due to exposure from routine agricultural application and subsequent transport to non-target areas, exposures can come from non-conventional uses of chemicals as well. The use of pesticides to harvest fish (henceforth "pesticide fishing") is one of these exposure routes. Pesticide fishing has been rarely described and never reviewed in the academic literature. The following review shows that it is common in much of the world's rural tropics.

### 1.1. Natural Poisons in Fishing

Natural poisons—typically derived directly from plants—have been used in fishing for millennia [12–14]. These practices are still common in some traditional communities today [15,16]. Natural fish poisons paralyze or kill fish and sometimes other aquatic organisms, and the practice has been banned in many contexts due to the ecological damage it can cause [13,16]. There is little to no research comparing the strength and toxicity of natural fish poisons to synthetic fish poisons. However, the impacts of natural poisons are likely less damaging than the use of synthetic chemicals, given that the natural chemicals are more easily metabolized in the environment (for example, pyrethroids such as cypermethrin persist much longer than pyrethrins, their natural counterparts which are found in pyrethrum flowers) [17]. The widespread availability of synthetic poisons in fishing communities that traditionally used natural poisons has lent to adoption of synthetic poisons in some contexts [15,18].

### 1.2. Pesticides and Other Synthetic Chemicals in Fishing

Researchers have only conducted a handful of studies on the use of synthetic chemicals in fishing, predominantly in a marine context. For example, cyanide fishing historically accounted for the majority of catch for the live reef fish trade (for aquaria and food) in the Indo-Pacific [19] and is still common practice [20]. Veitayaki et al. [15] review the use of destructive fishing practices for 23 Pacific island countries and report the use of pesticides (in five countries), bleach (in five countries), and cyanide (in two countries) in fishing to catch reef fish, as well as freshwater eels and prawns. In a survey in Ghana, 36% and 26% of interviewees described DDT and cyanide (respectively) as being used in marine fishing expeditions [21].

In the context of freshwater ecosystems, Von Brandt's classic book on fish-catching methods of the world mentions the use of quicklime, carbide, copper sulphate, and bleach in freshwater fishing [13]. More recently, Enuoh [22] reported the use of Gammalin 20 (active ingredient lindane)—an agricultural pesticide known to be toxic to fish [23]—to be highly prevalent in and damaging to freshwater fisheries in and around Cross River National Park, Nigeria. This fishery has been studied since the 1960s and fishermen had historically used natural ichthyotoxins [18]. Victor and Ogbeibu [24] monitored changes in macroinvertebrate species composition for months after an event where Nigerian fishers had applied Gammalin 20 to harvest fish in a stream pool that they had isolated by constructing a temporary mud dam. In El Yunque National Forest, Puerto Rico, residents illegally bleached a stream to catch freshwater shrimp within a well-studied forest, and scientists documented the case by monitoring the ecological succession thereafter [25]. In the Himalayan region of India one review describes the destructive use of lime and bleach, and the pesticides Nuvan, Thiodon, and Malathion, in remote mountain fisheries [26]. Additional studies from India, Nepal, Iran, Lake Victoria, Nigeria, Madagascar, Hawaii, Ecuador, and Costa Rica simply mention the occurrence of fishing with synthetic chemicals, but do not go into detail on the extent of its use or its impacts [27–38]. There are also some reports of the practice in the news from countries throughout the tropics [39–41].

Yet to date, no studies have done an in-depth assessment of the risks and extent of pesticide fishing in a freshwater system. International attention has focused more on large-scale illegal ocean fishing acts [42], but many of the impacts of small-scale illegal and destructive fishing practices remain poorly understood and unaddressed despite their potential to cause significant harm to the health of local people and ecosystems, especially to communities with subsistence based livelihoods.

### 1.3. The Context of Fishing with Pesticides in Southeast Nicaragua

Risk assessment unique to species and environmental conditions in Latin American countries are generally lacking, which limits the ability to effectively mitigate impacts from pesticides to aquatic ecosystems in the region [43]. In general in Nicaragua, human poisoning from pesticide use has occurred at a high rate for decades [44], affecting both pesticide users and their families [45], though it has been consistently underreported [46]. High levels of organochlorine pesticides have been reported in sediment, fish, and clam tissue in lakes and coastal areas, but similar studies for other kinds of pesticides and from river environments are lacking [47,48]. Incidents of fishing with pesticides in Nicaragua have never been reported in the scientific literature and rarely in the news. Yet based on observations from the field the practice is common and likely has wide-ranging effects on public health and aquatic ecosystem conservation. A 2017 news article told the story of some men who were almost killed from drinking the water of a recently poisoned creek [41]. Another recent story in the news described that a young man died and his brother was in serious condition after consuming a dead fish from a river—although the case was not confirmed as a poisoning the symptoms reported were characteristic of pesticide poisoning [49]. The Indigenous Rama and Afrodescendent Kriol communities who have legal rights to much of the land and resources of southeastern Nicaragua (see below) are greatly concerned about the issue and leaders have spoken out against it since 2016 [50]. The practice is illegal according to Nicaraguan National Law Number 489, Articles 35, 94, 123, and 125, and is punishable by a USD 5000 fine or 6−12 months in prison [51], and when carried out in national protected areas and Indigenous territories it is illegal under several more laws [52,53]. Yet incidents of fishing with pesticides have not once been processed in a court of law, even when they have occurred in protected areas and territories, due to poor surveillance and lack of political will. Recent legal complaints by Rama and Kriol forest rangers have highlighted the need for national support to address the issue [50,54].

### 1.4. Study Area: The Indio Maíz Biological Reserve

The 3158 km$^2$ Indio Maíz Biological Reserve is one of the largest five remaining tracts of primary rainforest in Central America. It is considered the core area of the UNESCO Río San Juan Biosphere

Reserve. Sixty-seven percent of the reserve overlaps with the Rama–Kriol Territory, comprised of 4842.56 km$^2$ of land and 4413.08 km$^2$ of sea that was jointly titled to the Rama and Kriol communities of southeastern Nicaragua in 2009. Indio Maíz is the only intact forest area of the terrestrial portion of their territory and extremely important for the subsistence livelihoods of traditional Rama and Kriol communities. In addition, it is a hotspot for populations of endangered and iconic animals such as the great green macaw (*Ara ambiguus*), jaguar (*Panthera onca*), and Baird's tapir (*Tapirus bairdii*), and hosts well-preserved river systems with intact fish and invertebrate communities [55,56]. Fishing with pesticides certainly threatens these preserved river ecosystems and the wildlife that depend on them. Each of the limited number of published studies from southeast Nicaragua calls for more research in these understudied and threatened systems [55–60]. Although previously not in the public eye, this reserve has been the focus of international attention in recent years due to the increasing extent of illegal deforestation, hunting, cattle ranching, and associated activity there. One example is the >5000-hectare fire of spring 2018, which was part of the motivation for the protests that fomented the current national political crisis [61].

### 1.5. Indigenous and Afrodescendant Peoples Confronting an Invasive Agricultural Frontier

Within southeast Nicaragua, long-settled communities of Indigenous and Afrodescendent peoples are losing land and ability to manage habitats to an advancing agricultural frontier driven by illegal settlers. There is a general movement of settler ranchers and farmers into eastern Nicaragua's rain-forested preserves from surrounding areas and drier western Nicaragua. Illegal settler agricultural activities along this frontier are the primary drivers of deforestation along Nicaragua's Caribbean coast. This is a major threat to the autonomy of Indigenous and Afrodescendant peoples in the region because it has dispossessed them of their ancestral lands and waters that have served as the bases of their subsistence and other economic activities [56,62,63]. It appears that the increased incidence of pesticide fishing in Indio Maíz and the Rama–Kriol Territory has coincided with the advance of the agricultural frontier.

The agricultural frontier represents an intense site for cultural–political contestation over the meanings humans make of the natural world in southeastern Nicaragua. Studying the cultural politics of water in the Florida Everglades (USA), anthropologist Jessica Cattelino argues that the study of cultural politics entails "attending to cultural practices like making meaning of nature, classifying it, and representing it, while also tracing how these cultural practices distribute resources among human groups and individuals" [64] (p. 238). In Indio Maíz, as in many other settler colonial settings like the Everglades, the most salient divide in these practices occurs between settlers compared to the Indigenous/Afrodescendant communities of the region. Rama and Kriol communities have historically lived from fishing, hunting, and agriculture over a wide range of their territory that allows for sustainable subsistence and adapts to the climatological setting [65–67]. By contrast, the *mestizo* settlers—of European, Indigenous, and/or African descent—have gradually occupied vast swathes of the territory for extensive cattle ranching and intensive agriculture that reflects a cultural politics in which nature is understood as requiring domination for proper use [66,68]. This stems from the ideological project of *mestizaje* [69–71] and government colonization and agrarian reform efforts over the past sixty years that promoted these practices [68,72,73]. This latter mode of cultural politics is reflected in the practice of fishing with pesticides insofar as it dominates the affected waterway and leads to immediate, extensive use without regard to long-term sustainability for those who live downstream from the site at or after the time of the use of pesticides. The boundaries between these two sets of practices are porous and do not neatly map onto the identities and community of "Rama" or "Kriol" on the one hand and "*mestizo* settler" on the other. That said, pesticide fishing fits within the cultural politics in terms of settler colonial logics of Indigenous dispossession and the domination of nature [64] (pp. 238–239), which makes this distinction between *mestizo* settler and Rama–Kriol salient for the purposes of this study.

*1.6. Study Objectives*

In order to confront this illegal invasion by settlers, Rama and Kriol communities have created an action plan for the protection and sustainable use of the portion of the Indio Maíz Biological Reserve that overlaps with their territory [74]. The plan includes the mitigation of illegal pesticide fishing as a major priority for the reserve, a process that begins with research and ranger patrols, and continues to legal action against environmental criminals and an educational campaign. This was the impetus for the study presented here.

The researchers involved in this project have several decades of combined experience in southeast Nicaragua and found through conversation with residents of the area that pesticide fishing is a major concern. Based on conversations in the field, a systematic study was developed in concert with the Rama–Kriol Territorial Government and three communal governments to better understand and address fishing with pesticides in and around the Indio Maíz Biological Reserve in southeast Nicaragua. The objectives of the study were first to review the literature on pesticide fishing, and second, to understand in the study context how pesticide fishing is carried out, what motivates it, its impacts on ecosystems and people, and how it might be prevented—using interviews, focus groups, and four years of data from forest ranger patrols. The team partnered with local people and organizations with the goal of using the resulting knowledge to inform the education and outreach efforts against the practice. The study was descriptive in nature since little is known about the practice.

## 2. Materials and Methods

*2.1. General Approach and Study Communities*

During February and March of 2019, 86 interviews and 5 focus groups were conducted in and around the Indio Maíz Biological Reserve (see Supplementary Materials, Section S6 for Interview Questions and Focus Group Activities). Two cultural/geographical groups participated in the study—(1) Communities legally residing within the core of the Indio-Maíz Biological Reserve (31 interviews, 4 focus groups), mostly made up of Indigenous Rama and Afrodescendant Kriol families who live within the core of the reserve and along the coast, as well as a few "old-stayer" *mestizo* families who have legal rights to live there and live similarly to the Rama and Kriol; and (2) cattle ranching and farming communities living along the agricultural frontier on the western and northern boundaries of Indio Maíz (55 interviews, 1 focus group), made up of *mestizo* families (Figure 1). Settlers who live in the reserve illegally were not included in interviews or focus groups. As described above, these communities have salient distinctions in their relationships with, and understandings of, the lands and waters around them. Most relevant for this study is that Rama–Kriol community members and the "old-stayer" *mestizo* families generally practice sustainable forms of agriculture, fishing, and hunting while *mestizo* settlers tend to engage in practices that cause deforestation, riverbank erosion, soil nutrient depletion, overfishing, and overhunting. Due to these general differences in agricultural, hunting, and fishing practices, many of the analyses in this study distinguish between the two forms of cultural politics generally found on one side or the other of the divide between communities residing legally in the reserve (mostly Rama–Kriol) and communities living along the agricultural frontier (*mestizo* settler communities).

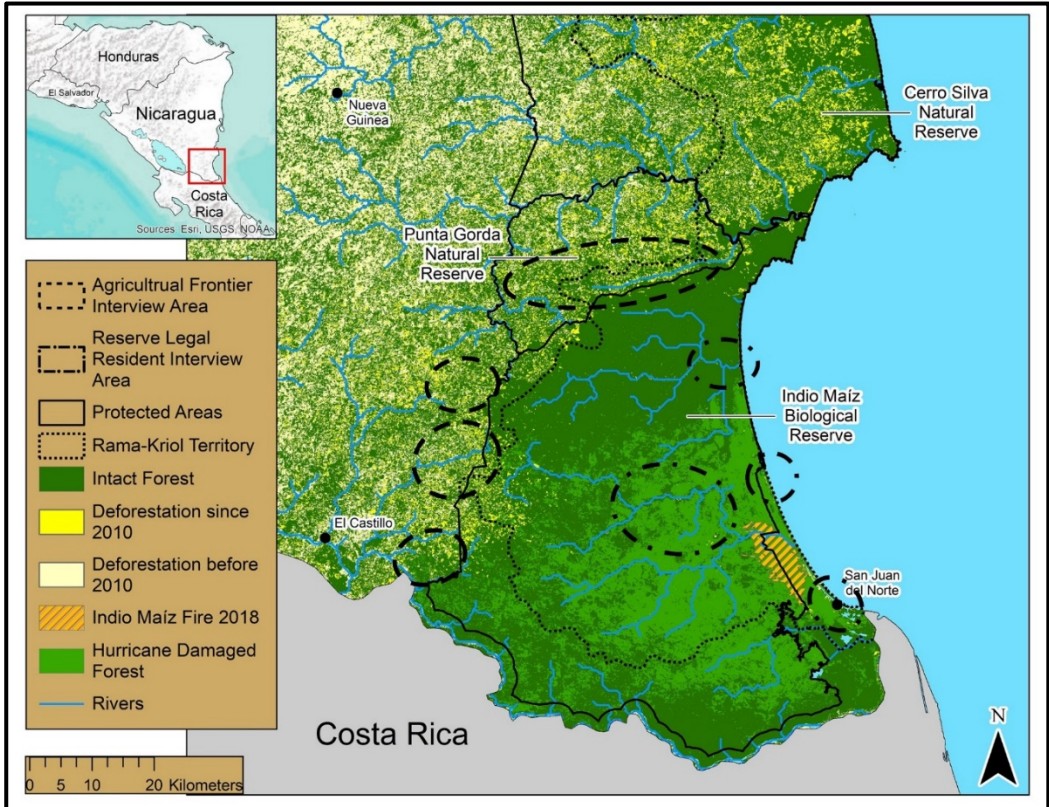

**Figure 1.** Map of the areas where the investigators conducted interviews in and around Indio Maíz, overlaid with forest loss (agricultural frontier) and hurricane damage data (Hurricane Otto, Nov. 2016) derived from the Global Forest Watch dataset (up to 2018) [75]. Includes the major fire from April 2018 derived from LANCE FIRMS NRT VIIRS 375 m Active Fire product [76]. Focus groups were carried out in each legal resident interview area and only in the furthest south frontier area.

## 2.2. Consent Process and IRB Approval

Interview and focus group questions were approved as exempt (category 2) through the Institutional Review Board at Michigan State University on 1/10/2019. Consent for interviews was obtained at the beginning of each interview after the reading of a consent script which explained the study objectives and process, and assured interviewees of anonymity and right to withdraw. Only interviewees who consented to audio recording were recorded (39 in total). Focus group participants were read a debriefing script at the beginning of each group, and anyone not in agreement with the terms did not participate. All focus groups were audio recorded. Because fishing with pesticides is illegal, interviewees were never asked directly if they have fished with pesticides, or the specific names of people who have both due to ethical concerns and for methodological purposes.

## 2.3. Sampling Protocol

### 2.3.1. Interviews

Interviews were carried out using an adapted snowball sampling technique instead of random sampling. This was chosen because the topic of pesticide fishing could be considered sensitive or hidden, due to its illegality and the stigma against it, and because it is not knowledge held equally across communities due to differing livelihood strategies and gender roles. Snowball sampling allows for a more targeted selection process and results in acquisition of more relevant information [77]. It was valid in this case because the purpose of the study was "primarily explorative, qualitative and descriptive" [77–79]. In cases of more concentrated communities (groups of families living closer

together), contact was typically first made with community leaders, elders, or religious leaders by a field assistant who knew them. In more dispersed communities (groups of families living further apart), individual houses were approached one-by-one. Recommendations for further interviewees in each community were provided by these leaders and/or the initial interviewee. Interviews were made with people who had spent at least 10 years living in a given community, and who had experience or knowledge of fishing in the area. When there were no more people in a community who met these pre-requisites, the team moved to another neighboring community. Typically, 5 to 10 interviews were carried out in each community.

A significant limitation of this study is the near lack of women in the sample of interviewees and a heavily male sample bias in the focus group component. This can be attributed to several factors in study design. In retrospect, the authors note a general tendency in the communities studied to direct contact with male outsiders to male heads of families upon initial contact. Establishing enough rapport for male researchers to be able to interview female participants can typically only be gained through deliberate efforts in longer term research. Given that research visits to each community were brief and all interviewers were men, this resulted in a sample that was almost exclusively male. This reflects a lack of consideration of gender in study design that should be corrected in future studies through longer term research and, most importantly, the inclusion of women and/or non-binary researchers in both the study design and research [80,81].

### 2.3.2. Focus Groups

At the time of the interview, all interviewees were invited to participate in focus groups, in cases where they were to be conducted in the community. Focus groups were held after any interviews were conducted in the community. Only in the community of Graytown was this not within 2 days of the last interview. A general invitation to participate in the focus group was also sent out by word of mouth to the community members. In many cases community leaders tried to invite adults with at least 10 years living in the area. Focus groups were made considering the dispersion of the houses and the number of possible interviewees in the community. Participation was typically between 10 and 15 people, and meetings typically lasted 1–2.5 h.

### 2.3.3. Ranger Patrols

The Rama and Kriol communities of Indio Maíz have three ranger teams that conduct monthly 5 to 10-day patrols within the area of overlap between Indio Maíz and the Rama and Kriol territory. Patrols are made by riverboat or canoe using rivers, lagoons, and streams and walking from the water body deep into the forest. While rangers patrol, they take evidence of any illegal human activity by taking pictures, notes, and marking GPS coordinates. Evidence of illegal fishing was noted by indirect observations (dead fishes, rock dams, pesticide containers, large fish smoking platforms, etc.). Methodology for patrols varied significantly depending on the objectives of the patrol. Patrols began in 2015 and data through 2019 was included. Patrol data are stored and analyzed in a Spatial Monitoring and Reporting Tool (SMART) database [82].

### 2.4. Data Analysis

Semi-structured interviews consisted of nine open-ended questions that guided the conversations. Question order varied between interviews. Interviewers encouraged further discussion of the themes using follow-up questions and other prompts. Interview questions can be found in the Supplementary Materials (Section S6). Not all questions were answered in all interviews, whether due to the interviewee choosing not to answer, the interviewee not having relevant information, or the interviewer omitting the question. These points of "no data" were taken into consideration in reporting in the following way: answers were categorized by response to a given question during the interview and reported as percent of interviewees out of the total for whom a response to that prompt was recorded (not including those points of "no data"). These distinctions in data processing are made clear in results reporting,

where for each question the number of interviewees with a given response (x) out of the total who responded to the question (y) is reported as n = x/y, and the number of non-respondents (z) is reported as no data = z. Although many of the questions may have been more suited for a survey technique, given the low literacy in the region, and the sensitivity and illegality of the study topic, interviews were used. This was seen as a better way to glean a more complete and representative sample.

Responses that could not be categorized (such as stories) were catalogued and reported as appropriate. No attempt at transcript processing was made. Interview and focus group participants were kept anonymous throughout the data processing. In many cases, responses were reported separately for the 31 (mostly Rama and Kriol) interviewees living within the reserve and the 55 interviewees living along the agricultural frontier (all *mestizo*) because differences in cultural politics and use of the practice between these communities were anticipated.

Focus groups consisted of an icebreaker and then three activities in which participants discussed (1) general challenges facing their community, particularly related to water resources, (2) pesticide fishing and its impacts on the community, and (3) actors implicated in fishing with pesticides, effects on these actors, and responsibilities they have or actions they could carry out to address it. These results contextualized the problem to each community and served as a productive participatory community conversation on the topic as a follow up to the individual interviews. Ranger data were used to augment the interviews and focus groups with further anecdotes and observations of the practice. Focus group activities can be found in the Supplementary Materials (Section S6).

## 3. Results

### 3.1. Interview Summaries and Anecdotes

#### 3.1.1. Frequency of Occurrence

When asked whether fishing with pesticides occurs in the rivers in their community, 87.5% of the interviewees living in the reserve (mostly Rama or Kriol residents) who explicitly answered this question said that it occurs in the river in their community (n = 21/24, no data = 7). Among interviewees along the agricultural frontier, 67.3% said that the practice occurs in the river in their community (n = 33/49, no data = 6). Some who said it did not occur in their community said they were aware of it happening elsewhere.

When asked how frequently people thought it occurred in their community, results indicate that it occurs commonly, especially in some communities. Responses varied from as little as "not for 8 years", to as frequently as 3–4 times a week. Considering all interviewees (from both along the frontier and in the reserve), 38 % (n = 19/50, no data = 36) indicated that it happens at least 3–4 times a year, 40% (n = 20/50, no data = 36) indicated that it occurs most in the dry season (January to May) when rivers are clearer with lower water levels, and 3 specified that it is most common during holy week—the week before Easter (1 ag. frontier and 2 Rama interviewees), since eating fish is a holy week tradition.

In some places along the agricultural frontier the practice is common still, but not as common as it was when the settlers had recently arrived to the area and rivers were still full of fish and shrimp (~10–30 years ago, depending on the community). For example, when asked about whether or not pesticide fishing occurs in his community, one interviewee stated, "To kill fish no, but to kill those little animals—shrimp, prawns, yes . . . like 15 days ago they said someone was poisoning Bartola River." After asking if they fished with pesticides as often as in the past, he said "since there aren't any left anymore, no, hardly not anymore, they've run out…They used to frequent that little creek over there and haul out a bucket-full of shrimp . . . every 15 days or so a different person would come by to fish that way there. All the little creeks used to have shrimp and prawns." Another interviewee stated it bluntly, "Well, nobody fishes with poison here [in his community creek] anymore simply because there aren't any shrimp left."

Interviewees living legally in the reserve, where there are still healthy rivers and abundant fish and shrimp, tended to report a higher rate of occurrence and were more likely to discuss the topic.

In the areas where these communities live, pesticide fishing, when it happens, is said to be carried out in their rivers by illegal *mestizo* settlers coming from outside. Only 25 of all 86 interviewed individuals responded to the prompt regarding how long ago a specific case that they knew of occurred. Of these, 12.0% (n = 3/25, no data = 61) said in the last 3 months, 16.0% (n = 4/25, no data = 61) said in the last year (but longer than 3 months), 32.0% (n = 8/25, no data = 61) said in the last 5 years (but not in the last year), and 40.0% (n = 10/25, no data = 61) said longer than 5 years ago. All of the known and reported cases by interviewees living within Indio Maíz were within the last 5 years (n = 8), half of these being in the last year, whereas along the agricultural frontier 41.1% were in the last 5 years (n = 7/17), 3 of these 7 cases being within the last year. (Note: these were only cases where interviewees mentioned time since a known case, and numbers do not represent the rate of pesticide fishing occurrence).

### 3.1.2. Target and Site Characteristics

Fish and crustaceans were reported as key targets of fishing with pesticides. Crustaceans, especially freshwater shrimps and prawns were reported as the target more frequently than fish, especially along the agricultural frontier, although both taxa groups were consistently represented. Of the 39 agricultural frontier interviewees who explicitly specified fishing target species, 20.5% (n = 8/39, no data = 16) mentioned both fish and crustaceans, 69.2% (n = 27/39, no data = 16) only mentioned crustaceans, and 10.3% (n = 4/39, no data = 16) only mentioned fish. Of the 22 interviewees living in the reserve who explicitly specified fishing target species, 45.5% (n = 10/22, no data = 9) mentioned both fish and crustaceans, 40.9% (n = 9/22, no data = 9) only mentioned crustaceans, and 13.6 % (n = 3/22, no data = 9) only mentioned fish. Of all 61 interviewees who specified fishing target species, 73.8% (n = 4/61, no data = 25) mentioned prawns (*Macrobrachium* spp.), 67.2 % (n = 41/61, no data = 25) mentioned shrimps (*Atya* spp.), and 27.9 % (n = 17/61, no data = 25) mentioned crabs (*Pseudothelphusidae* spp.). Specific fish mentioned as targets were machaca (*Brycon guatemalensis*), guapote (*Parachromis* spp.), sardines (*Astyanax* spp.), snook (*Centropomus spp.*), and small cichlids (such as *Amphilophus* sp., *Amatitlania* spp., *Crybroheros* spp., and *Tomocichla tuba*).

Only 52.3% of all interviewees (n = 45/86) responded to the prompt for characteristics of the fishing site. Of the 20 interviewees living in the reserve who specified this information, the most common responses were that people target drier shallow areas, pools or slower water, or headwaters or small streams (45.0% (n = 9/20), 35.0% (n = 7/20), and 15.0% (n = 3/20), respectively; no data = 11). Of the 25 agricultural frontier interviewees who specified, most commonly reported were that people target areas where they expect to see fish or shrimp, headwaters/or small streams, in areas with more current, drier shallow areas, and pools or slower water (35.0% (n = 9/25), 28.0% (n = 7/25), 16.0% (n = 4/25), 12.0% (n = 3/25), and 12.0% (n = 3/25), respectively; no data = 30). Rocky areas, creek mouths, and areas with high forest cover were also mentioned. Those who said that predominantly fish were targeted mainly mentioned slow water and pools (and stream mouths), while those who targeted headwaters, shallower and rockier areas, and faster current identified the practice as primarily targeting crustaceans, although both fish and crustaceans were reported as being targeted in both habitat types.

### 3.1.3. Pesticide Fishing Process

Over 75% of all interviewees (n = 65/86) specified the chemicals used in pesticide fishing. Cypermethrin and Deltamethrin were the most commonly reported, followed by Aluminum Phosphide, Dichlorvos, Methomyl, Glyphosate, and Carbofuran (Table 1). Other chemicals mentioned by interviewees included "Herbicides" in general, agricultural lime, and diesel fuel. One person said that "bombs" have been used in the past. Agricultural frontier interviewees tended to report a higher diversity of chemicals. The potential effects of these chemicals to human health and ecosystems according to the literature are reported in Table 1. Six interviewees living in the reserve and three from along the agricultural frontier mentioned that plants are used or have been used to catch crustaceans or fish. Seven species were reported and are listed in Table 2.

**Table 1.** Agricultural pesticides reported as used in fishing, with information about their impacts on human health and ecosystems and their intended use. For most chemicals, severe skin exposure can result in symptoms like those of ingestion/inhalation. Information from PubChem Hazardous Substances Data Bank [83].

| Chemical | Cypermethrin | Deltamethrin | Aluminum Phosphide | Dichlorvos | Methomyl | Glyphosate | Carbofuran |
|---|---|---|---|---|---|---|---|
| **Name used for product by interviewees** | Cipermetrina, Supermetrina | Butox, Decis | Pastilla del amor, Pastilla para curar frijoles | Torsasem, Torvan | Nudrin | Glifosato, Round-up | Furadan |
| **% reported use in fishing** | 70.8% (n = 46/65, no data = 21) | 35.4% (n = 23/65, no data = 21) [1] | 9.2% (n = 6/65, no data = 21) [2] | 7.7% (n = 5/65, no data = 21) [3] | 6.2% (n = 4/65, no data = 21) | 4.6% (n = 3/65, no data = 21) | 1.5% (1 person) |
| **% only in past** | 0.0% | 15.4% | 1.5% | 3.1% | 0.0% | 0.0% | 0.0% |
| **Class of Chemical** | Pyrethroid insecticide /acaricide | Pyrethroid insecticide | Phosphide insecticide /rodenticide | Organophosphate insecticide/acaricide | Carbamate insecticide/acaricide/ nematicide | Phosphano-glycine herbicide | Carbamate insecticide/acaricide/ nematicide |
| **Human symptoms: Skin/eye exposure** | Redness, burning or tingling sensation, numbness, itching | Redness, burning or tingling sensation, numbness, itching, watering of the eyes | Redness, burning sensation. | Irritation, redness, blurred vision, absorbed into skin (see Ingestion /Inhalation) | Redness in eyes, blurred vision | Eye/skin irritation, redness, eye damage | Irritation, absorbed into skin (see Ingestion /Inhalation) |
| **Human symptoms: Ingestion/ Inhalation** | Cough, shortness of breath, dizziness, headache, nausea, vomiting, convulsions | Numbness of tongue and lips, salivation, cough, sore throat, abdominal pain, vomiting, muscle twitching, convulsions, unconsciousness, death | Headache, dizziness, sore throat, cough, shortness of breath, nausea, vomiting, diarrhea, abdominal pain, convulsions, shock or collapse, unconsciousness, death. Reacts with water and stomach acids to produce toxic phosphine gas | Headache, dizziness, chest tightness, wheezing, sweating, ataxia, low blood pressure, salivation and nasal mucous, bluish discoloration of the skin, nausea, vomiting, diarrhea, muscle twitching, convulsions, respiratory paralysis, death | Headache, dizziness, sweating, difficulty breathing, salivation, diarrhea, nausea, vomiting, weakness, muscle twitching and cramping, convulsions, unconsciousness, death | Cough, burning sensation in the throat and chest, convulsions, coma | Headache, dizziness, incoordination, blurred vision, sweating, difficulty breathing, salivation, nausea, vomiting, diarrhea, muscle twitching, weakness, convulsions, unconsciousness, death |
| **Human symptoms: Chronic exposure** | May cause effects on nervous system, possible human carcinogen | May cause effects on nervous system, not likely to be carcinogenic to humans | Not a carcinogen | Dermatitis, nervous system damage, possible human carcinogen | May cause anemia, kidney, or liver damage, not a carcinogen | Debated as possible human carcinogen | May cause effects on nervous system, not likely to be carcinogenic to humans |
| **Ecosystem effects** | Very toxic to fish and aquatic invertebrates (acute and chronic) | Very toxic to fish and aquatic invertebrates (acute and chronic) | Turns into phosphine gas—very toxic to all animal life (acute), can be absorbed by gills or skin of fish or invertebrates and cause death | Very toxic to birds, animals, fish, and aquatic invertebrates (acute and chronic) | Very toxic to fish and aquatic invertebrates (acute and chronic) | Can be toxic to fish and aquatic invertebrates, especially formulations with certain common surfactants | Very toxic to fish and aquatic invertebrates (acute and chronic) |

**Table 1.** *Cont.*

| Chemical | Cypermethrin | Deltamethrin | Aluminum Phosphide | Dichlorvos | Methomyl | Glyphosate | Carbofuran |
|---|---|---|---|---|---|---|---|
| Biocon-centration | High Potential | High Potential | None | Low potential | Low potential | Low potential | High Potential |
| Intended Use | Crop insect pest control, household insect pest control, eliminate livestock parasites | Crop insect pest control, household insect pest control, eliminate livestock parasites | Crop fumigant (especially for storage), rodenticide | Crop fumigant (for storage and in field application), eliminate livestock parasites | Crop multi-pest protection, household insect pest control, eliminate livestock parasites | Broad-spectrum herbicide to kill weeds or terminate crop | Crop multi-pest protection, possibly other uses |

[1] 10 of these 23 interviewees described Deltamethrin as only being used in the past. [2] 1 of these 6 interviewees described Aluminum Phosphide as only being used in the past. [3] 2 of these 5 interviewees described Dichlorvos as only being used in the past.

**Table 2.** Different plants which were reported as being used for their natural poisonous properties to catch fish.

| Scientific Name | *Pentaclethra Macroloba* | *Hura Crepitans* | *Hyeronima Alchorneoides* | *Anacardium Excelsum* | *Mucuna sp.* | *Entada Gigas* | *Serjania Mexicana* |
|---|---|---|---|---|---|---|---|
| Local Name | Gavilán | Jobillo | Nanciton | Espavel | Ojo de Buey | Escalera de mico | "Cola de iguana" |
| Plant Type | Tree | Tree | Tree | Tree | Vine | Vine | Vine |
| Times reported | 3 | 1 | 1 | 1 | 1 | 2 | 1 |

Only 37.2% of all interviewees (n = 32/86) specified the form in which pesticides are applied. Of those, 71.8% (n = 23/32, no data = 54) said that liquid formulations are dropped in the water, while 28.1% (n = 9/32, no data = 54) said that powdered (or liquid) formulations are mixed with flour, typically from corn, and that is dropped in water. For each chemical, it was not always clear whether it was principally used in liquid or powdered form. Eighteen of the twenty-three interviewees talking about using liquid pesticide said it is used for crustaceans, while only four of these said that liquid pesticide is used for fish. Seven of the nine interviewees who described the practice of mixing pesticides with corn flour say this method is used for fish, while only one said it was used for crustaceans. Some interviewees specified that it is put in the water by hand (dripped from a container), others said it is dispersed using a rod or paddle, or even shirts soaked in the chemical. One interviewee specified that first corn flour without the poison is thrown in the water, then once the fish are primed, the flour with poison mixed in is thrown in. Aluminum phosphide was associated with the corn flour method.

Collection method of poisoned organisms was specified by 38.4% interviewees (n = 33/86). Fish and shrimp collection by hand was most commonly reported (78.8%, n = 26/33, no data = 53), whereas four Rama–Kriol interviewees mentioned that they had heard of collection using a rock dam created downstream. Spears, nets, diving, and naturally occurring vines were also mentioned. Two interviewees mentioned putting soap in the water in order to stop the chemical from continuing to affect stream life. Another interviewee mentioned that there was a "counter" method to deactivate the chemical but did not describe it.

### 3.1.4. Perceived Use and Motivations

Of the 31 interviewees living in the reserve, 75.0% (n = 15/20, no data = 11) believed that the fish and shrimp caught using pesticides are sold, 45.0% (n = 9/20, no data = 11) believe it is for subsistence, and 35.0% (n = 7/20, no data = 11) believe that it is used as bait for fish in bigger rivers. On the contrary, only one interviewee from the agricultural frontier mentioned that the catch is sold, whereas 86.0% (n = 37/43, no data = 12) said that it is for subsistence, and 37.2% (n = 16/43, no data = 12) said that it is used as bait for fish in bigger rivers.

Interviewees living in the reserve and along agricultural frontier perceived the similar motivations people have for fishing with pesticides. Of the 72 who responded to the prompt of what motivates someone to fish with pesticides compared to other methods, 66.7% (n = 48/72, no data = 14) said people fish this way because it is easier to catch fish and shrimp, 26.4% (n = 19/72, no data = 14) said because one can catch more fish, 16.7% (n = 12/72, no data = 14) said because one can catch fish faster, and 13.9% (n = 10/72, no data = 14) said that it is a good or the best way to catch bait (small shrimp). Other respondents speculated that others fished with pesticides because it was tradition, that people were bored and it was a vice, and that it was the most economical option available for fishing (fishing line is too expensive).

### 3.1.5. Effects on Humans and Livestock

Interviewees living in the reserve and along agricultural frontier were concerned about impacts of fishing with pesticides on human health (69.1%, n = 47/68, no data = 18). A total of 24 individuals specified what specific health impacts they had witnessed or heard about. Regarding cases of human consumption of water from poisoned rivers, participants reported stomach issues or diarrhea (n = 7), cough (n = 2), and even convulsions (n = 1). Concerning cases of skin or eye exposure to contaminated water, participants reported blindness (n = 2) and skin infection/rash or burning sensations (n = 3). Some interviewees mentioned concerns about long-term impacts from exposure, such as cancer (n = 7). Six interviewees specifically mentioned effects on pregnant women (miscarriage). Nine interviewees (five reserve residents and four ag. frontier) were concerned about impacts to human health via impacts to food security, stating that fishing this way leaves the rivers without fish for the community in the future. Agricultural frontier interviewees were particularly concerned about impacts to health of livestock (57.1%; n = 24/42, no data = 13), 23 of these mentioning livestock miscarriages due to drinking

contaminated water. Many of these impacts were reported as firsthand experiences after people or livestock were knowingly exposed to river water contaminated via fishing with pesticides. People confirmed firsthand experiences by describing having seen shrimp, fish, or insects jumping from the water, swimming irregularly, or coming upon a stream with fish and shrimp dead on the shore, and in some cases from physical symptoms like feeling or smelling the chemical in the water as it passed (skin irritation or chemical odor).

An interviewee who lives near the Agua Zarca Creek along the agricultural frontier described that people who bathe in the river can develop a rash if exposed to pesticide contaminated water. He described a case from about 20 years ago where many people were poisoned from chemicals in their river, some of whom had to be hospitalized. According to participants in the focus group in Corn River (within the reserve), there was a case in 2018 where it was clear that a main tributary to the river had been poisoned. There was large mass of small fish and shrimp that floated downstream. They explained that during the following days children and several adults from the community located at the mouth of Corn River became sick with diarrhea and two cows owned by one of the participants of the focus group miscarried their calves. People were concerned that even their wells were not safe. Another example of concern for impacts to human health became apparent during the focus group with the community of Indian River. They discussed that there are seasons of the year (after the floods and in the dry season) in which they get sick to the stomach and do not know what is the cause, but suspect that it is related to their drinking water, which comes from the main channel of the river. Many community members suspect increasing contamination from upstream as the reason for this discomfort. This could be related to other public health concerns as well but given that these downstream communities drink directly from rivers, exposure to chemicals from pesticide fishing upstream is a risk.

### 3.1.6. Perceived Ecosystem Effects

People perceive that the chemicals used in fishing are more deadly to crustaceans than to fish. When asked what species or group is most affected, 73.1% of all interviewees (n = 49/67, no data = 19) mentioned crustaceans, while only 34.3% (n = 23/67, no data = 19) mentioned fish. Clams were also mentioned as affected. Impacts to other aquatic life (stream insects, algae, etc.) were rarely mentioned—only one interviewee said that it also affects "other species" but was not specific. Another said that the poison also kills the streamside grasses. One specified that with low doses, sometimes fish are not killed and can be revived. Multiple interviewees said that smaller fish are affected more than bigger fish, and for a longer distance downstream. When asked how the fish and shrimp respond to the poison, it was common for people to say that the organisms go blind or become "stupid" or disoriented. Other common responses were that the shrimp jump out of the water to and try to escape to the stream edge (and are thus easy to collect), that the fish float or flip upside down, and that they run out of oxygen.

When questioned about potential impacts to wildlife, 90% (n = 54/60, no data = 26), mentioned that they think it impacts wildlife negatively, whether directly via consumption of poisoned water (mentioning death, miscarriage, general health decline), or depletion of resources upon which wildlife depends. Caiman (*Caiman crocodilus*) and Neotropical River Otters (*Lontra longicaudis*) were mentioned as specific examples of the latter.

When asked if fish return to a stream after poisoning, 22% of all interviewees responded no (n = 8/36, no data = 50), while 77.8% responded yes (n = 28/36, no data = 50). When asked if prawns return to a stream after poisoning, 22.7% responded no (n = 10/44, no data = 42), 77.2% responded yes (n = 34/44, no data = 42). Answers of how long it takes for fish and shrimp to return to a poisoned stretch of river varied from within a week to 5 years or longer. Four interviewees specified that they come back the first time the river has high flow again, which lines up with case studies of disturbance response in similarly flashy tropical streams, as long as there is a source population and the organism drifts or is mobile [25].

When asked how far downriver the effect persists, 16.7% of interviewees (n = 8/48, no data = 38) said less than or equal to 50 m, 25% (n = 12/48, no data = 38) said between 50 and 300 m, 10.4% (n = 5/48, no data = 38) said between 300 m and 1 km, 8.3% (n = 4/48, no data = 38) said between 1 and 2 km, 18.8% (n = 9/48, no data = 38) said between 2 and 10 km, 12.5% (n = 6/48, no data = 38) said more than 10 km, and 8.3% (n = 4/48, no data = 38) said all the way to the stream mouth. Thus, responses varied but more than 50% believe that the chemical stops affecting the stream after 1 km. Some qualified that it depends on the concentration which is put in the stream as well as the stream size and flow.

When asked how the river water changes when poisoned, 42.6% of all interviewees (n = 26/61, no data = 25) reported a change in color to a milky, white, or yellow color, 36.1% (n = 22/61, no data = 25) reported a distinct smell, 8.2% (n = 5/61, no data = 25) reported color change to dark or murky, 4.9 (n = 3/61, no data = 25) said that bubbles form, and 27.9% (n = 17/61, no data = 25) said that nothing changes. Individuals reported specific effects, for example, that deltamethrin (Butox) makes the water white and smelly, that Dichlorvos gives the water a greenish hue, or that Glyphosate makes the water "oily". The bubbling and distinct smells reported are likely associated with use of aluminum phosphide, which turns into toxic phosphine gas upon interacting with water, when most of the substance bubbles out and some becomes dissolved in the flow. Clearly, visual physical effects in the water depend on the chemical used.

### 3.1.7. Perceptions of Legality and Guardianship

When asked if the practice is illegal, all who answered said yes (n = 77/77, no data = 9). Only 2 interviewees answered that they knew what specific laws made it illegal, while 71 said that they did not (no data = 13). When asked if they thought they were national, territorial, or local laws, answers differed between study communities. Interviewees living in the reserve and along agricultural frontier thought it was illegal at the national level, but only interviewees living in the reserve (mostly Rama or Kriol) mentioned laws that make it illegal at the local or territorial level. This shows a lack of awareness of regulations based on Nicaragua's Law 28, the autonomy law, in the agricultural frontier community. When asked about the legal consequences of being caught fishing with pesticides, 90.9% of interviewees who answered (n = 50/55, no data = 31) said they thought one would be imprisoned, and 29.1% (n = 16/55, no data = 31) thought one would be ticketed (both of which are true legal consequences under the national law number 489).

When asked if there is any local guardianship on this issue, 51.4% of interviewees who answered (n = 37/72, no data = 14) said that there is. Twenty-one interviewees mentioned local restrictions by farm owners who restrict access to rivers on their claimed land, 8 said that the practice is prohibited at the communal level, and 4 mentioned that they have received education efforts against it. Five (from within the reserve) mentioned that the Rama–Kriol forest rangers gather information and file cases against it. Two mentioned that they heard on the radio that the practice was bad, and two mentioned that MARENA (the national environmental authority) used to address the issue, but that they had stopped doing so. No one mentioned a case where legal action has been taken by authorities against someone fishing with pesticides. One agricultural frontier interviewee stated it simply, saying "It's illegal . . . but I don't think it's prohibited, because they do it a lot . . . nobody says anything. There is no defense for the animals."

### 3.2. Focus Group Activities

### 3.2.1. Community Concerns

The first focus group activity identified several challenges faced by community members. In the communities that held a focus group within the reserve (n = 4 communities), problems identified were lack of political will for conservation (n = 4), declining fish and wildlife populations or illegal/over-harvest of wild sources of food (n = 4), poor government regulation of natural resources (n = 3), lack of army/national park service surveillance/checkpoints (n = 3), lack of health care

(n = 3), invasion of communal lands by settlers (n = 2), lack of employment opportunities (n = 2), poor transportation (n = 2), lack of electricity or fuel (n = 2), lack of sanitation (n = 2), and lack of education (n = 2). Other issues mentioned by just one focus group were contamination of rivers, lack of community planning, danger from settlers, danger from wildlife, crop damage by wildlife, natural disasters (like hurricanes), exclusion from of state development projects, and disrespect for autonomy laws and Rama cultural heritage. In the one agricultural frontier community that held a focus group, problems identified were lack of employment, poor or eroded soils, eroded trails or roadways, changes in climate, and low water flows (see Supplementary Materials, Table S1 for these results split by focus group community).

### 3.2.2. Impacts of Pesticide Fishing

The second activity identified the main impacts of fishing with pesticides. Between the five focus groups, these included reductions of fish and shrimp (food insecurity) (n = 4), poisoning of people (n = 3), cattle miscarriages/negative effects to livestock (n = 3), contamination of rivers (n = 2), loss of cultural values/heritage (n = 2), loss of tourism potential (n = 2), reduced job opportunities (n = 1), and effects to trees (n = 1) (see Supplementary Materials, Table S2 for these results split by focus group).

### 3.2.3. Actors, Effects, and Actions

The third activity identified specific actors connected to pesticide fishing, negative and positive outcomes that the activity results in for those actors, and actions that they could pursue to address the impacts of the practice. These results are outlined in Table 3. The actions proposed by the focus groups generally include increasing application of the law, increasing surveillance, decreasing access to pesticides, promoting education and outreach, building community capacity, and supporting co-management strategies.

**Table 3.** Actors implicated in fishing with pesticides, the effects of the practice on them—positive (+) or negative (−), and actions these actors could take to address the issue, as defined by the 5 focus groups.

| Actors | Effects | Actions |
|---|---|---|
| Rama and Kriol Community members | (−) Lose food source, health risk, lose opportunities for work (fishing, tourism), and lose livestock and associated earnings | Issue public legal complaints to institutions, invest in legal capacity of communities, increase surveillance (ranger program), prosecute/remove perpetrators, make an educational campaign, talk to neighbors, hold meetings with local government, advocate use of co-management agreement with the state, better management of transportation and work allowed in the reserve, tax fishers and fish purchasers, and invest in protection of the forest |
| Rama and Kriol Community Leaders | (−) Lose time to affect change, lose credibility with community | |
| Pesticide using fishermen | (+) Make money; (−) Lose future fishing/earning prospects | Stop fishing with pesticides, fish other ways in permitted areas, follow the law, respect legal fishing boundaries, and pay taxes to the Rama and Kriol communities |
| Nicaraguan authorities (municipal and state) | (+) Short term keeping of support/popularity (by not punishing), make money; (−) Long term losing of support (Increasing degradation of resources), and losing credibility/respect with community | Increase regulations and supervision for fishing (and other extractive activities), regulate and supervise pesticide vendors, apply the law, increase surveillance (activate ranger program), establish control points in the reserve, protect of the forest, and take back up the co-management agreement with the communities |
| Pesticide vendors and producers | (+) Make money; (−) Risk losing license or product sustainability | Stop selling chemicals near Indio-Maíz, and only sell to crop-livestock producers who properly use chemicals (certification system) |
| Fish purchasers (at the market) | (+) Make money | Stop purchasing fish from Indio Maíz, and pay taxes to the community |
| NGOs | (−) Lose opportunities for projects | Support outreach/educational campaign and community ranger programs |

### 3.3. Ranger Patrol Results

In addition to collecting a variety of other data on the illegal invasion of Indio Maíz (new farms, houses, roads, deforestation, etc.), rangers reported 54 possible incidents of illegal fishing (possibly with pesticides) during the years of 2015–2019 (Figure 2). In one case in April 2019, near the mouth of the Pimienta River in Indian River, an abnormal amount of dead fish (species *Tomocichla tuba*) were observed floating downstream. This was likely caused by a fish poisoning event upstream.

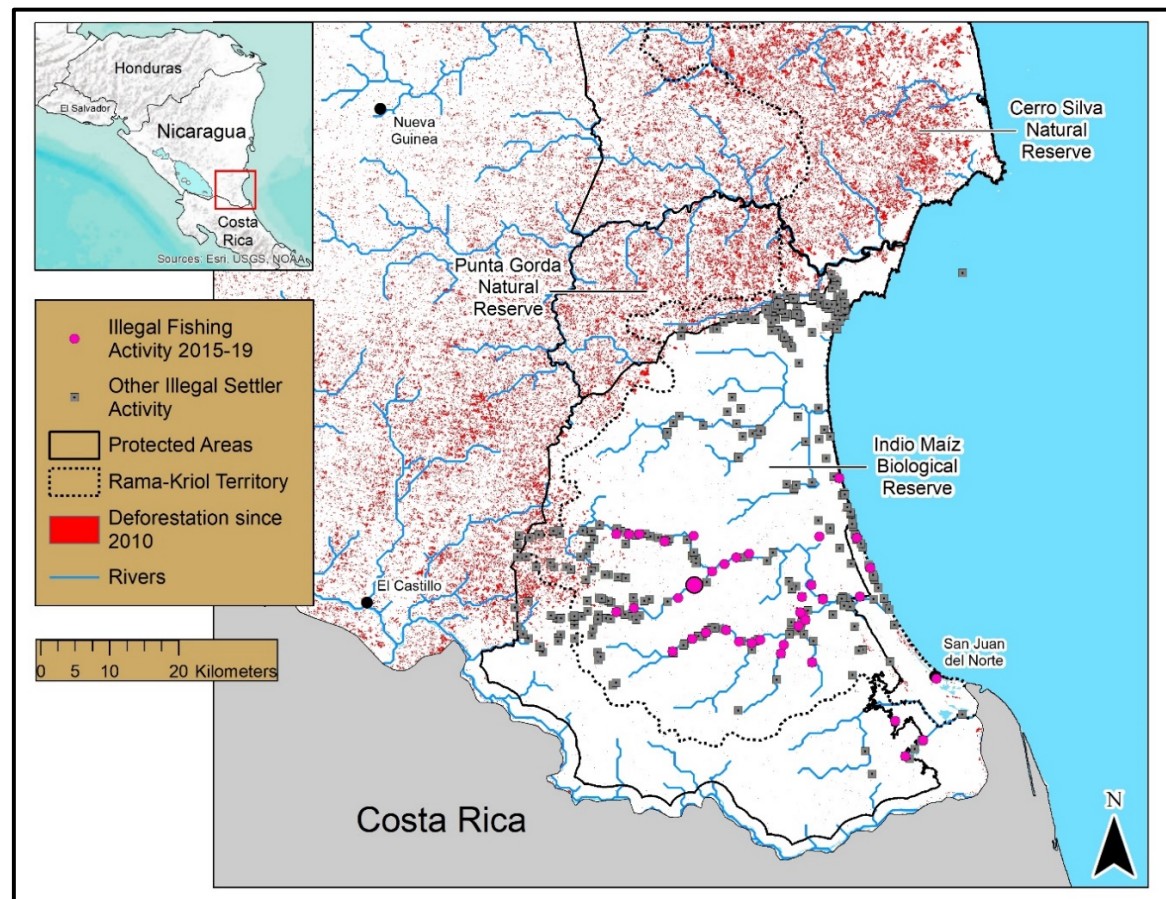

**Figure 2.** Map of illegal settler activities reported by the rangers in Indio Maíz from 2015 to 2019, highlighting cases of illegal fishing. Pink points are cases of illegal fishing (possibly with pesticides) in the study period (2015–19), the larger pink point is a very likely case of fishing with pesticides (mentioned in-text). The remaining grey points are other cases of illegal settler activity from 2015 to 2019 including deforestation, wood extraction, roads, crops, livestock, camps, and houses from settlers recently invading Indio Maíz. Overlaid with recent deforestation (agricultural frontier) derived from the Global Forest Watch dataset [75].

In none of the patrols were the rangers able to confirm people actively using pesticides for fishing. The cases of illegal fishing that were reported were mostly abandoned riverside camps with fish or game smoking platforms, and in some cases rock dams made to capture fish. Fishing practices used at these abandoned camps are unknown, but interview results indicate that use of pesticides is likely, especially where rock dams were found, and given the large size of the drying racks at many of the camps. One example was a case in 2014, when community members were travelling up Indian River to fish, and came across a team of nine mules at the riverbank, and signs of a large group of men accompanying them. Wanting to avoid a violent confrontation, they turned the boats downstream and paddled the 2-day trip back to get the support of the rangers. When they arrived days later, they found hundreds of fish heads and spines, a smoking rack, and a rock dam. It was likely in that case that the

men had been illegally poisoning and harvesting fish—they suspected that the rock dam was to catch the poisoned, dying fish, and the mules were to bring smoked fish to markets in the agricultural frontier (Figure 3). In this case, most of the fish remains were of the Bobo Mullet (*Joturus pichardi*), which is a species of conservation concern in Costa Rica, and of unknown conservation status in Nicaragua [84]. Residents said that it took years for the fishery at this site to recover.

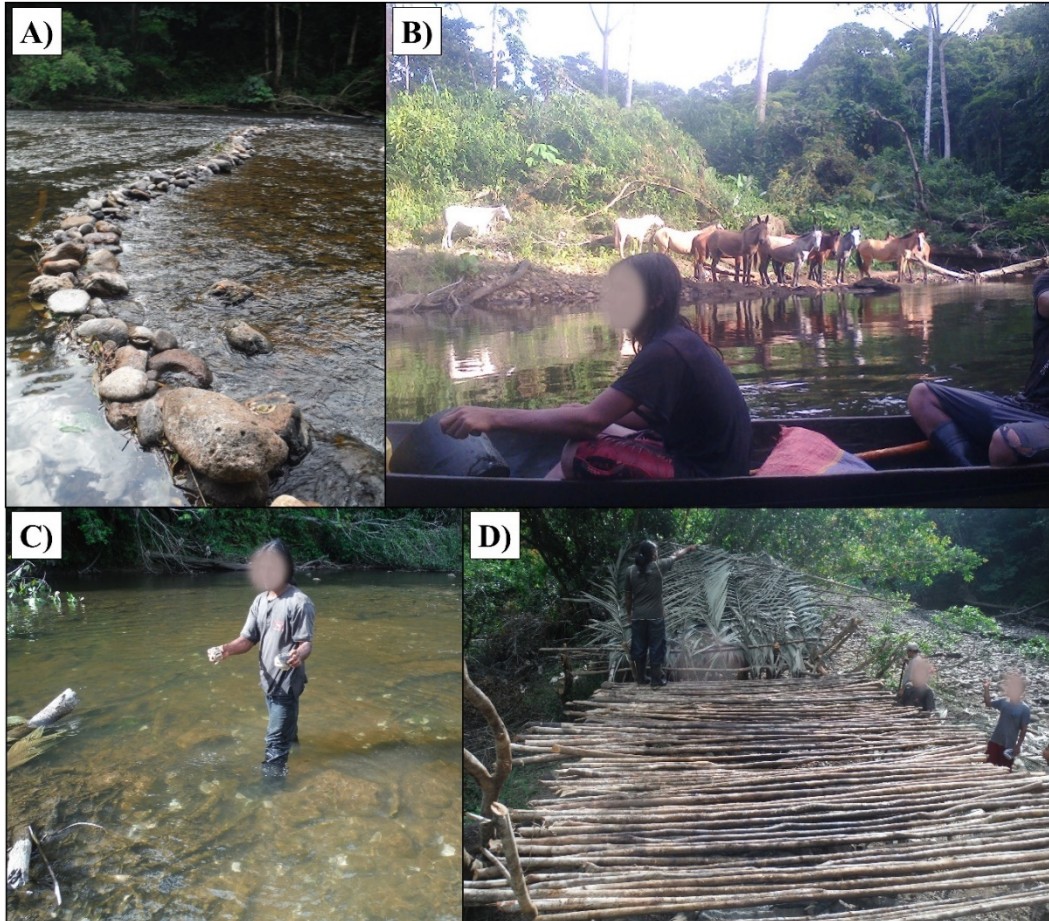

**Figure 3.** (**A**) A rock dam created to trap (likely poisoned) fish. (**B**) A team of mules encountered by Rama community members deep in their territory up Indian River, 2014. (**C**) Bobo Mullet (*Joturus pichardi*) fish heads scattered across the river bottom (white spots underwater). (**D**) A giant smoke-rack, for smoking large quantities of (likely poisoned) fish, next to a temporary lean-to camp. *Photo credit:* Rama Forest Rangers.

Due to the limited geographic scope of the patrols, the massive extent of the reserve, and the illicit (and thus hidden) nature pesticide fishing, encounters of direct evidence of fishing with pesticides proved difficult for rangers. Thus, some forest rangers were included in the interview and focus group conversations as well. This allowed for cases they heard by word of mouth during patrols to be included in the study. In interviews with the rangers, they reported a higher incidence of the practice than was reported by the average interviewee. This is likely since these individuals spend more time out in the reserve than most community members and have more incidence with illegal activities.

## 4. Discussion

Given the high prevalence of the practice of fishing with pesticides found in this study, the practice has clearly been under-reported and unaddressed in southeast Nicaragua. It is happening at a high rate within the agricultural frontier along the border of the Indio Maíz Biological Reserve. It is

also happening deep within the reserve, near traditional Rama and Kriol communities, perpetrated presumably by settlers who have illegally invaded the area. According to interviewees and ranger data, the practice appears to be most common in pristine rivers and creeks, and decreasingly so as rivers become more depleted of organisms. This is alarming, given that this reserve holds some of the last well-preserved aquatic ecosystems in the country and is a core area of cultural heritage for traditional communities with river-based subsistence livelihoods. The chemicals reported and the means that are used to catch fish are a cause of great concern.

### 4.1. A Real Ecological Threat

Cypermethrin and deltamethrin, both pyrethroid insecticides, were the most reported chemicals used for fishing and are known to be extremely toxic to fish, and even more toxic to aquatic insects and freshwater shrimp [85–87]. These chemicals are lipophilic, so when they hit water they are quick to adhere to the gills and tissues of fish and other organisms [87], as well and to the suspended organic matter in water or in sediments where benthic macroinvertebrates are found and readily absorb the chemical [88]. The pesticides properties database [89] describes lethal concentrations for cypermethrin to be very low—0.00021 mg/L for aquatic invertebrates (48 h $EC_{50}$), 0.0128 mg/L for aquatic crustaceans (96 h $LC_{50}$), and 0.00151 mg/L for fish (96 h $LC_{50}$)—levels which would be far exceeded in the case of intentional pouring of pesticides into water for fishing. Cypermethrin takes a week to two months to degrade in rivers, which is enough for the chemical to affect aquatic life for kilometers downstream (half-life of 5 days [90]; half-life of 13–50 days [91]). This is especially concerning because concentrations as low as <0.004 µg/L have been shown to have sublethal effects on fish such as disruption of reproductive function and thus population health [92]. Considering the toxicity of these and other chemicals reported as used it can be inferred that the effects of pesticide fishing on freshwater biodiversity are likely to be substantial, as much or even more than from more typical (non-fishing) routes of exposure because of the comparatively high concentrations likely used in pesticide fishing [3–5]. Reported impacts from the interviews and focus groups such as strong reactions of fish and shrimp to poisons and loss of stream life for up to kilometers downstream of a poisoning site corroborate that these severe ecosystem impacts are occurring.

Given these known and reported impacts, river ecosystems in the Indio Maíz Biological reserve are surely being negatively affected by pesticide fishing [4,5]. A concurrent study [56] documented declines of fish, shrimp, and aquatic invertebrate diversity and abundance because of deforestation for cattle ranching along the agricultural frontier, when compared to pristine streams in Indio Maíz. Although some sites differed between this and the concurrent study [56], given the extent to which fishing with pesticides was said to occur in the region, it is likely that some portion of these declines could be attributable to the impacts of fishing with pesticides.

In addition, Indio Maíz is a sanctuary for rare wildlife, and has been identified as a core habitat for the endangered Baird's Tapir, which is declining globally due to similar agricultural frontiers [93]. There may be significant direct risk posed by pesticide fishing to tapirs, who live their lives centered around rivers and streams in Indio Maíz [94]. Other river-associated wildlife are also at risk.

### 4.2. Threats to Human Health

According to the literature, human health impacts from exposure to cypermethrin and deltamethrin are serious as well. The chemicals can cause irritation, burning, and numbness with skin exposure; gastrointestinal and respiratory distress, convulsions, and even paralysis or death with high levels of exposure, and could lead to cancer (cypermethrin) and nervous system degradation, or even impacts on the immune system and metabolism (see Table 1) [83,95]. In other contexts, pesticide residuals (including cypermethrin) have been detected in breast milk [95,96]. Depending on the chemical used in fishing, the threats it poses to ecosystem and human health differ (See Table 1), as do its persistence in sediments and fish tissue [83]. However, with nearly all chemicals mentioned in this study, not just cypermethrin and deltamethrin, the risks to human health can be severe. Severe impacts like those that

would be expected based on the chemicals used were reported by interviewees as related to pesticide fishing incidents: cases of chemical burns and eye damage from skin exposure to poisoned river water; cases of stomach issues, diarrhea, cough, convulsions, and even miscarriage from ingestion of poisoned river water; and miscarriage in livestock that ingested poisoned river water.

Communities along the agricultural frontier and in the forested reserve are especially at risk because rivers and streams are used for drinking water, bathing, washing clothes, transportation, and livestock watering (livestock legally outside the reserve). This daily use of river water creates a real risk of instantaneous exposure to high levels of chemical in the case of water use directly downstream of a pesticide fishing event. It also creates the possibility of complications from chronic exposure [83,89], especially for people further downstream where the chemicals could be continuously present due to intermittent use in fishing throughout the headwater areas. This could disproportionately affect downstream residents—whom, in the case of Indio Maíz, are predominantly Indigenous Rama and Afrodescendant Kriol communities (the agricultural frontier is in the headwaters). Some pyrethroids like cypermethrin have been found to accumulate in fish tissue [83,97], which may provide another route of exposure for river residents. Unfortunately, chemical assays of the water, sediment, fish tissue, or human tissues to search for these substances or their degradation products were not carried out in this study. Such studies will be important in future research to determine the actual extent of exposure risk to people and animals.

Direct exposure of fishermen to the pesticides they use is also very concerning. For example, most of the aluminum phosphide applied to water bubbles out as toxic phosphine gas, which is surely inhaled by the fisherman during fishing. Moreover, given that there is poor pesticide handling education in these communities, skin exposure during fishing is likely. One interviewee reported a case where a teenage girl lost vision in one eye because she got a pesticide in her eye while fishing. It is also very likely that some exposure to chemicals occurs when fish captured this way are consumed, especially for cypermethrin and deltamethrin which are lipophilic and adhere to animal tissue. This is alarming given that most of the agricultural frontier interviewees (86%) specified that fish and shrimp are captured for consumption in the home.

The observation that most interviewees living in the reserve (75%) reported was that pesticide fishers sell their catch, while only one of 54 agricultural frontier interviewees reported that catch is sold, has a few different potential interpretations: (1) that the people living in the reserve (mostly Rama and Kriol) only thought it to be sold, while in reality it is not, (2) that people along the frontier are unwilling to admit that poisoned fish are being sold, or (3) that catch from within Indio Maíz is brought out for sale, while catch in the frontier communities is more commonly for consumption or bait. Given the snowball sampling method, another explanation could be that a bias towards interviewees with similar/shared knowledge of cases of sale existed. The second and third interpretations (with potential for snowball sampling bias) seem most likely when also considering that the rangers have found at least 10 large meat drying racks at sites deep in the reserve where illegal settlers have presumably harvested and smoked large quantities of fish and wild game to bring outside of the reserve. Given that it is likely that at least some sale of catch is occurring, the possibility of unknown consumption of pesticide contaminated fish by purchasers is another potential exposure route, and a reason for concern.

### 4.3. Problems of Land Governance

The issue of pesticide fishing is situated within the broader problem of the agricultural frontier—the illegal invasion and usurpation of land in the forested reserves and Indigenous/Afrodescendant territories of Nicaragua. Not only are municipal and national governments systematically failing to address illegal land grabbing and deforestation, but they are also failing to address the outright illegal activities like pesticide fishing that are being carried out by these settlers [98]. When asked about enforcement, not one interviewee mentioned a case where legal action has been taken against someone fishing with pesticides. Only the surveillance efforts of local farm owners and the Rama-Kriol forest rangers were mentioned, yet these rangers are not deputized and thus do not have legal right to apply

the laws against fishing with pesticides or detain those doing it. Given the clear results of this study, and the many calls by Rama and Kriol communal governments to authorities to do something about the issue [98], it is past time that the relevant government authorities respond to pesticide fishing with increased surveillance and legal action in the context of a broader approach to protecting the ancestral lands and ensuring the food security and safety of the Indigenous and Afrodescendant population of Indio Maíz.

### 4.4. Conservation Action and Future Research

This study supplies a novel and detailed understanding of how pesticide fishing is carried out. This information is valuable to any authority or organization who may want to implement measures to prevent this environmental crime, in Nicaragua and anywhere the practice occurs. The prevalence of the practice demands targeted action and education. Recommendations from the focus groups for preventing this environmental crime for relevant stakeholders are outlined in the focus group results (Table 3). The Rama–Kriol Action Plan for Indio Maíz specifically calls to (1) organize ranger patrols and investigations to find out about pesticide fishing, (2) prosecute known poisoners, and (3) develop a high-profile public awareness campaign against fish poisoning [74]. The first of these objectives was partly achieved by this study, although continued targeted ranger patrols are necessary and further investigation would be helpful. The second of these objectives is the responsibility of territorial, regional, and national government entities, and is critically important. The third objective was carried out in 2019, and was informed by a number of realizations from this research: (1) that community members and the fisheries they depend on are at risk because of pesticide fishing—particularly the Indigenous and Afrodescendant communities that live downstream from the agricultural frontier, (2) that many people are unaware of the extent to which fishing with pesticides can damage human health and the sustainability of river fisheries, (3) that those who are aware but still carry out the practice clearly have a myopic view of river resources and community health, which is unsustainable and unethical, (4) that there is poor awareness of the laws surrounding pesticide fishing, and (5) that community members are currently the only effective form of guardianship against the practice. Given these findings, an educational poster, YouTube video, and radio ads were developed and disseminated along the agricultural frontier, on the radio, and on social media (Supplementary Materials; Figure S3, Multimedias S4 and S5a,b). Local communities and universities held press releases and presentations on the issue, reaching thousands of local residents [99]. The campaign explained the risks of the practice, explained the legality, asked people to report known cases, and called for a moral stand against it—encouraging local people to call out neighbors for fishing with pesticides in rivers near where they reside. Andriamalala et al. [36] found a social marketing campaign approach to be effective at reducing incidence of destructive fishing practices like poison fishing in Madagascar. The campaign in Nicaragua that followed this research is another example of an impactful approach, though it is unknown if it resulted in lower incidence of pesticide fishing. The authors hope that it that it will be replicated in other contexts.

Given that illegal settlers in Indigenous lands are presumably responsible for this destructive practice that poses significant health risks to the land's rightful owners, the Indigenous Rama and Afrodescendant Kriol, and that it threatens their food security and autonomy, it is also critical for the Nicaraguan State to address this by removing these settlers from Indigenous lands in Indio Maíz. This approach will need to entail improved law enforcement in Indio Maíz to control future invasion, sustainable livelihood programs for farmers along the border of the Rama and Kriol Indigenous territory that disincentives land invasion, and targeted development programs for illegal settler families relocated from within Indigenous lands. Continued capacity building for Rama and Kriol communities living in the reserve to address these cases will also be important. At the current time, the clear neglect of the Nicaraguan State to assist in the protection of Indio Maíz at even a level basic enough to guarantee the food security and cultural survival of the reserve's Indigenous and Afrodescendant populations must change to ensure a healthy future for the reserve and its inhabitants.

## 5. Conclusions

In the absence of systematic effort to address environmental crime in Nicaragua, the agricultural frontier will continue to move into the Rama and Kriol Territory and illegal pesticide fishing and forest loss will likely increase significantly. The long-term health of regionally important river ecosystems and the food security, autonomy, cultures, and health of Indigenous and Afrodescendant populations supported by those ecosystems may be lost if nothing is done by the Nicaraguan State and the international community to support local efforts. While there is systematic impunity for environmental crimes in Nicaragua, the practice of fishing with pesticides is not unique to Indio Maíz. This novel understanding of the practice in one context may provide insight to the issue in other geographies and cultural contexts. Given the mentions of the practice in the primary and grey literature from sites around the world, it seems likely that fishing with pesticides is common and under-reported many remote communities. Pesticides are readily available in most rural communities and using them to catch fish can lead to high fishing success rates. As in the case of southeast Nicaragua, fishing with pesticides could be affecting rural communities and ecosystems wherever pesticides are used. Therefore, the practice likely poses a threat to inland fisheries and human health that is global in scale and requires attention on a global level. Sustainability of inland fisheries has recently been emphasized as of utmost importance in achieving the United Nations Sustainable Development Goals, particularly No Poverty (SDG 1), Zero Hunger (SDG 2), Clean Water and Sanitation (SDG 6), Responsible Consumption and Production (SDG 12) and Life on Land (SDG 15) [8,100]. The human health consequences of pesticide fishing also raise concerns for achieving Good Health and Well-being (SDG 3). More resources must be devoted to addressing this issue both in Southeast Nicaragua and in other remote sites around the world. Further research must work wherever this occurs to understand why it is occurring and to what extent, and to quantify exposure levels and risks to human and ecosystem health.

**Supplementary Materials:** The following are available online at http://www.mdpi.com/2071-1050/12/23/10152/s1, Table S1: Challenges faced by community members that were identified in each focus group, Table S2: Negative impacts of pesticide fishing mentioned in each focus group; Figure S3: An educational poster that was developed in light of this research; Multimedia S4: An educational video that was developed in light of this research; Multimedia S5a,b: Radio ads that were developed in light of this research; Additional information S6: Interview Questions and Focus Group Activities.

**Author Contributions:** Conceptualization, J.T.B., C.A.J., and A.J.D.; methodology, J.L.M., J.F.M.E., A.J.D., J.T.B., and G.R.U.; data analysis, J.T.B.; investigation (field work), A.J.D., J.F.M.E.; data curation, J.F.M.E., A.J.D.; writing—original draft preparation, J.T.B.; writing—review and editing, J.T.B., J.L.M., C.A.J., J.F.M.E., A.J.D., and G.R.U.; visualization, J.T.B.; supervision, G.R.U., C.A.J.; project administration, J.T.B., C.A.J., and A.J.D.; funding acquisition, C.A.J. and J.T.B. All authors have read and agreed to the published version of the manuscript.

**Funding:** This research was funded by the New England Biolabs Foundation, 2018 grant "Ending illegal fishing with pesticides in the Indio Maíz Biological Reserve". Author J.T.B. was supported by US Student Fulbright grant during part of his work on the project. The APC was funded by author G.U., with research funds from Lyman Briggs College, Michigan State University.

**Acknowledgments:** Thanks to Alejandro Mairena, Marcial Tenorio, Orlando Joseph, Erick Thomas, Cristobal Duarte, Erick Duarte and Luis Altamirano Lumbí for their assistance in the field work. Thanks to many people in the communities of Sumu kat, Corn River, Indian River, Greytown, and Diriangen for sharing their stories before the formal interview process, which were crucial in designing the questions used in the interview and focus group processes. And thanks over 100 individuals in these and other communities for their participation in the research. Special thanks to the Rama–Kriol Territorial Government and the directive boards of communities of Greytown, Indian River, and Corn River, who consented to and supported this study and follow up actions on their communal lands and in their communities. Thanks to the Waiku *Centro de Arte y Diseño* and *Comunicaciones Sanles Alemán* for their design and production support with outreach materials (Supplementary Materials S3–S5).

**Conflicts of Interest:** The authors declare no conflict of interest. The funders had no role in the design of the study; in the collection, analyses, or interpretation of data; in the writing of the manuscript, or in the decision to publish the results.

**Ethics Statement:** Ethics approval for this study was granted through the Institutional Review Board at Michigan State University as exempt (category 2) on 1/10/2019.

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
