# Peer review of "Fishing with Pesticides Affects River Fisheries and Community Health in the Indio Maíz Biological Reserve, Nicaragua"

_sustainability, doi:10.3390/su122310152_

Round 1

Reviewer 1 Report

Dear authors, here is my opinion about the article entitled “Fishing with pesticides affects river fisheries and community health in the Indio Maíz Biological Reserve, Nicaragua”, submitted for consideration for publication in Sustainability.

The study deals with a very sensitive subject, of which many communities are not aware of the harmful effects that pesticide fishing can have on their health and on the health status of their environment. This descriptive study, based on interviews and focus groups, was carried out in a rigorous manner; the manuscript is very well structured and the obtained results show that local authorities in each country where this illegal approach is known, should be concerned and pay more attention to raising awareness and punishing people who do not respect the law that prohibits this fishing practice.

Since this study is purely descriptive, direct conclusions such as the one reported in the summary (lines 30-31) should be avoided.

According to the results of the interviews, several pesticides were used by the fishermen. Cypermethrin and Deltamethrin were the most commonly reported pesticides for this illegal practice. In my opinion, it would have been very relevant, and would have given even more weight to this interesting study, to make chemical assays in the water to search for these substances or their degradation products, especially in the areas downstream of the discharge points. This approach could support the discussion point that was addressed in the discussion section of the manuscript (lines 654-656).

Such field studies should be supported, and applied on a large scale in areas where this kind of practice is still tolerated despite its prohibition by law. Studies based on interviews and focus groups combined with further investigations to really assess the state of health of the environment (Ecotoxicological studies), as well as the state of health of the individuals living in this environment (man, fauna and flora), should be encouraged in the future.

Hence, I strongly recommend this article for publication in Sustainability journal, but some points (see below) should be taken into account before that.

Comments:

Lines 30-31: You cannot be really sure and report that these symptoms are due to pesticides fishing practice. This conclusion should be mentioned in this way, only if a cause and effect relationship has been established.

Line 51: please provide the full meaning of the abbreviation used at its first mention

Lines 50-54: Note that, even if these new pesticides are supposed to be less persistent in the natural environment, because they are considered as degradable compounds, many of them are often detected in river waters in the form of residues (i.e. chlorpyrifos, parathion, mecrocpro, etc.). These residues show the opposite concentration trend in surface water (increasing concentrations over the years) which can pose a real long-term health problem (see review of Fernando P. Carvalho, 2017. https://doi.org/10.1002/fes3.108)

Line 126: Replace “KM2” by “km2

Line 219: please provide the full meaning of the abbreviation used at its first mention

Lines 654-656: It is not the objective of the study, but it would have been very relevant to look for these chemicals or their degradation products in water, upstream and downstream of the discharge points.

Author Response

Thank you for your helpful comments and suggestions! I attached a document with an itemized response to all of your concerns. Additionally, you may consult the track changes document to see where changes were made.

Joel Betts

Reviewer 2 Report

The manuscript collects information on the practice of pesticide fishing in Nicaragua mainly based on interviews from the local population. This is an important topic, which is interestingly wrapped in this manuscript but is currently not present enough in science and global media. Based on this, this publication is likely read by scientists from different professions, which is also reflected in the selection of the journal.

Currently, however, the manuscript is in sections hard to understand for environmental scientists, like me, who are unfamiliar with terms of the humanities. For terms such as “community”, the definitions or current usage needs to be clarified to avoid confusions whether biological or human communities are meant.

Moreover, the results gained from interviews and focus groups cannot be presented as hard facts as well as observations by rangers are only indications for the use of pesticide finishing, but no proof. Additionally, the results might be biased due to the use of the snowball technique. All these uncertainties and possible biases need to be clearly expressed throughout the manuscript, especially in the results, discussion and conclusion sections, which are likely to be read more frequently. In the introduction and materials sections, it is written quite clearly that this is mainly an explorative study. The sudden jump to present everything as hard facts is a clear overselling of the study. Alterations to present the necessary uncertainty should not be to much work, since most times adding / altering a single work should be sufficient.

In the results section, some sentences are hard to read and understand due to the high number of referred numbers. Most of these sentences could be formulated a bit more elegant, by e.g. adding most of the numbers in brackets.

  1. Detailed comments:
  • Lines 7-14: For affiliations 1,2,4 and 5 mentionings of the county (USA) is missing.
  • Line 21/23: The term “buffer” in this context not clear. Consider either defining buffer or use the term “catchment” since this is clear.
  • Line 22-23: Maybe improve / alter this sentence. Not clear which community the forest rangers refer to in this context. See general comments to the manuscript.
  • Line 27: Improve this sentence. Maybe to “The use of ichthyotoxic woody plants were more rarely reported.” Since this is a more traditional way of catching fish, this information should be added and clarified why this is important enough to be mentioned in the abstract.
  • Line 33: Define or explain “Rama” and “Kriol”. For me, who is unfamiliar with the study region, these terms are completely unclear.
  • Line 43: “gamut” is a not that common term in this context. Consider altering this word to also increase readability for non-native English speakers.
  • Line 43 - Reference 3: Consider adding / using a more up-to-date reference.
  • Line 46 – Reference 5: Consider exchanging this reference with Schäfer 2019. This is a more up to date reference and in contrast to the previously cited reference (5) a review comprising results from several studies, reference 5 included.
  • Line 56: Consider changing “spillover” to another term, since transport to non-target areas can also happen differently (e.g. runoff, leaching, spray drift…). Spillover might be one of the minor pathways in agriculture.
  • Line 58: Change to “exposure routes”.
  • Line 83: Consider changing “copper vitriol” to “copper sulphate” since the first term is outdated and not frequently used.
  • Line 87: Maybe add a reference to section 1.1.
  • Line 90-95: Maybe shorten these sentences and focus on the important message to the present manuscript.
  • Line 94: What is an experimental forest?
  • Line 115: See above. Maybe improve the definition? For readers not familiar with the study regions, the names “Rama” and “Kriol” could also refer to villages.
  • Lines 117 – 121: Maybe sum this section up a bit by using the first sentence and then shortly refer to even stricter rules in national protected areas and indigenous territories.
  • Line 118: Is this amount US $? Is this amount written in the law or is the amount converted to (US ?) $?
  • Lines 121 – 123: If available, maybe add rationale. Where the people doing this not caught, were charges filed against someone not followed?
  • Line 126: Change “KM²” to “km²”
  • Line 129: Use “km²” instead of “square kilometers” as above or change the unit in line 126.
  • Line 134: What are “intact fish and shrimp migrations”? Maybe better refer to communities, even though the communities might migrate or specify. It is a pity that both references are not easily accessible once due to language and once since referring to a Master’s thesis. Consider adding a link to the pdf of the Master’s thesis in the reference section.
  • Line 139: Threat to what?
  • Line 143: Consider mentioning the study region again.
  • Line 143: Here “indigenous” is written with a lowercase letter, in contrast to most parts of the manuscript. Please check the whole manuscript for consistency.
  • Line 146: Here it is written “illegal settlers”, while in the abstract (and once in the discussion) you referred to “illegal squatters”. To increase readability and understandability, consider using the same word throughout the manuscript. I would recommend the use of “settlers”.
  • Line 155: Maybe add the information in which country the Florida Everglades are located, even though it should be widely known.
  • Line 172: In contrast to the rest of the manuscript “Rama” and “Kriol” is written in quotation marks and mestizo not in italics. Please alter this to be consistent in the manuscript.
  • Lines 177 – 192: Please expand this section. It would be important that it also includes information which methods over which extent was used to conduct this “systematic study”. It would also be important to mention the study region / country in this section since this is often the most read section in introductions and it is common to have a very short (1-2 sentences) overview of the methods.
  • Line 189: Here a definition or specification of “communities” is important. For me (an environmental scientist) this would clearly refer to biological communities such as fish, invertebrates etc. For humanists, this might mean something different.
  • Line 195: Maybe define “focus groups” since this might not be clear to readers from other fields (like me) or refer to section 2.3.2.
  • Line 195 – 212: Maybe add the rationale why the number of focus groups differed hugely between the two study groups.
  • Line 213: Figure 1: Consider changing the colour palette. For colour-blind readers, green and red are not distinguishable and therefore the map is not readable. Please also consider using a different colour palette for “Indio Maiz Fire”, since the same colours are used for the deforestation. Also, here “Maiz” is written with a normal “i” in contract to the rest of the manuscript. I would recommend being consistent in the whole manuscript.
  • Line 238, 240, 244: It would help readers which are not experienced in this field, that “communities” would be defined better. Especially, to discriminate it from the biological term. Maybe “village” would be better understandable for a wider readership (if correct)?
  • Line 245-254: Could the limited number of women in the study also partially be explained by fewer women actively involved in fishing?
  • Lines 277-278: Please refer to the section in the SI, where the interview can be found.
  • Line 301-303: To call the ranger observations “hard data” is a huge overstatement! Please rephrase.
  • Line 318: Were these three interviewees from the same community? It should be specified either way.
  • Line 321: When did the settlers recently arrive in the area? This information would help to estimate the effect on the biological communities.
  • Line 333: Please specify from how many communities/interview areas these interviewees were. Maybe the people were more open / honest in one of the regions.
  • Lines 353-357: The names here and in the questionnaire in the SI differ. Please unify them.
  • Line 355: Please change “Pseudothelphusidae” to italics.
  • Lines 372 – 374: Did the interviewees used the names of the active ingredients (it sounds like it) or the name of the formulations?
  • Line 376-377: Please do not enter discussion to the results section.
  • Line 377: Could this difference also be related to the pesticides known to the respective group of interviewees?
  • Line 382 – 386: Table 1:
    • Maybe it would be helpful for the reader of also include “type” (insecticide, herbicide) in this table.
    • “Local names”: Are these formulations or local names for the active ingredients? If first, please alter the row name.
    • “External exposure”: What do you mean with “irritation”? Mental irritation or irritation of skin etc.?
    • “intended use” Consider specifying “crop protection”. Are not all pesticides designed for one or another way of crop protection? Maybe call it crop protection from insects?
  • Line 391: Why different form to present the number of interviewees?
  • Line 391-392: Better write: “Of those 71.8% (n=…”
  • Line 392: Are all of the above-mentioned formulations liquid? If not, this sentence would need to be rewritten.
  • Lines 415-424: These sentences are mostly discussing the results. Please move this section to the discussion.
  • Lines 415-420: Another option would also be the sampling method. Using a snowball sampling it is likely that people with similar knowledge are interviewed.
  • Line 421: Maybe add “presumably” since drying was most likely not observed.
  • Lines 422-424: This is not really a summary, since this was not mentioned before.
  • Line 435: Maybe better “groups of interviewees”?
  • Line 448: Is it known, how the interviewees knew that the water was contaminated?
  • Line 453: Is it correct to write “focal group” in this context?
  • Line 479-481: Why report both results? Only mentioning that 90% of interviewees assumed a negative effect on wildlife is sufficient.
  • Line 490-492: This is again more discussion. Consider shifting this to the discussion section.
  • Line 493-497: Is there maybe a connection of living / fishing region of the interviewees and their answers?
  • Line 505: Why using the name of the active ingredient in the case of Glyphosate, but not for the other pesticides?
  • Line 511: Why is the n not reported as a ratio here?
  • Line 514: Same as the comment of line 435.
  • Line 519: I would recommend rephrasing “of all interviewees” since this ratio only refers to the ratio of interviewees who answered this question.
  • Line 522: See above. I would recommend not to use “of all interviewees”.
  • Lines 533ff: Why this distinct separation of focus groups, when some results were already presented in the sections before? Why is the number of communities written with a capital N?
  • Line 534-548: What is with the focus group which was held in the agricultural frontier?
  • Line 550: Move or remove “(negative)”.
  • Line 563-565: Table 3: To equalize pesticide using fisherman with illegal settlers is based on an assumption. Since the interviewees were not asked if they used this technique themselves, it cannot be illuminated that they also used this technique. Better at least add “presumably”.
  • Line 568: I would recommend adding uncertainty to this sentence. Eg. “rangers reported 54 possible incidents of illegal fishing”.
  • Lines 572-580: Figure 2: Same comments than above for Figure 1. Additionally: Pink and purple are barely distinguishable. Maybe consider using different shapes. “illegal activity” and “illegal settler activity” does not necessarily mean the same. Please unify. “confirmed case of fishing with pesticides (mentioned intext)”: I haven’t read anything about any confirmed cases, only observations which are likely to be explained by pesticide fishing. See also general comments to the manuscript.
  • Line 583: But it could also be likely that they 1) used another fishing technique (e.g. dynamite fishing, electric fishing – I know this is unlikely but possible) or 2) dried game using these platforms. Uncertainty needs to be written more clearly.
  • Line 588-591: See above. It is very likely, but not clear.
  • Line 610: See above. It is very likely, but not proven / clear.
  • Lines 613-614: See above. It is likely done by the illegal settlers not this is not proven or observed.
  • Line 620: Consider changing to “ecosystems”.
  • Line 621: This is not clear. Maybe other pesticides/insecticides were used but the interviewees are unaware of them. Rephrase.
  • Line 626: Use subscript “50” for “LC50”.
  • Lines 626-628: Consider using a checked source for LC50 values, such as the Pesticide Properties DataBase https://sitem.herts.ac.uk/aeru/ppdb/en/Reports/197.htm).
  • Line 633-636: This sentence has a misleading conclusion. The effects are not there because the taxa are infrequently monitored. The effects are just not traceable. The part “their contributions to ecosystem health underappreciated” is also incorrect, at least globally. I would recommend entering citations of reviews or studies focusing on effects of insecticides on stream communities like in the introduction.
  • Lines 637-649: Add references on effects of pesticides on humans. Not only by the one database cited in Table 1 but also from studies.
  • Lines 654-655: Consider using the official term, chronic, for “long-term low concentration”.
  • Lines 650-669: Here citations are missing. Consider also adding information that pesticide residuals were detected in e.g. breast milk (Bedi et al, 2013; DOI: 10.1016/j.scitotenv.2013.06.066) or other tissues (Genuis et al., 2016; DOI: https://doi.org/10.1155/2016/1624643). Both studies are not from the study region, but similar exposures and accumulations could occur.
  • Line 686-687: Enter a reference proving the effects of pesticides on stream ecosystems /aquatic life.
  • Lines 685-696: It is not clear, why this section is separated from “Real risks to ecosystem and human health”. Please consider having one section on ecological effects including results from the same study area and one on effects on human health.
  • Lines 687, 690: Consider writing “a previous study” instead of “Betts”. Consider adding a link to the pdf of the Master’s thesis in the reference section.
  • Line 689: Consider altering this sentence to include that this study is purely based on interviewees and not on observations. Maybe the interviewees were not correctly informed or they purposely did not tell the truth.
  • Line 725: It is already clear that this approach was successful?
  • Lines 727 – 728: It is likely, but not clear that the illegal settlers using this technique. Please rephrase this sentence. Please also consider using “settlers” instead of “squatters” (see above).
  • Line 739-754: Consider including an outlook to the conclusion section. This could include investigating effects on stream communities, pesticide analysis in the streams or detected fish residuals to gain information on the real used pesticides and the analysis of human tissue to estimate effects on humans.
  • Line 741-742: The increase in illegal fishing and forest loss is likely but not clear. Please include the uncertainty to this sentence.
  • Supplementary information
    • Page 5: Multimedia S4: Isn’t WhatsApp a form of social media?
    • Page 6: Sentence: “In practice, Spanish translations of interviews questions and activities were used in most cases.” Is sadly not understandable. Please rephrase
    • Page 6: Please add scientific names to the animals which were part of question 1. Since the names were translated from indigenous languages / Kreol to Spanish and from Spanish to English, it might without the pictures hard to know which animals were part of your study. This is maybe not important for this manuscript but for the one, which is additionally prepared.

Author Response

Thank you for your very detailed reading and response to this manuscript. You have helped us improve it immensely. We appreciate your many helpful comments and suggestions! I attached a document with an itemized response to all of your concerns. Additionally, you may consult the track changes document to see where changes were made.

Joel Betts

Reviewer 3 Report

Refer to attached Word document for comments and revisions

Author Response

Thank you for your detailed reading and response to this manuscript. We appreciate your many helpful comments and suggestions! I attached a document with an itemized response to all of your concerns. Additionally, you may consult the track changes document to see where changes were made.

Joel Betts

Round 2

Reviewer 2 Report

The comments by both reviewers were incorporated into the manuscript in a very good way.  No further questions arose and thus the manuscript is now suitable for publication.

Author Response

Thank you for your favorable reply to my responses to your concerns!

Regards,

Joel Betts
